# A Workflow for Computer-Aided Evaluation of Keloid Based on Laser Speckle Contrast Imaging and Deep Learning

**DOI:** 10.3390/jpm12060981

**Published:** 2022-06-16

**Authors:** Shuo Li, He Wang, Yiding Xiao, Mingzi Zhang, Nanze Yu, Ang Zeng, Xiaojun Wang

**Affiliations:** 1Department of Plastic Surgery, Peking Union Medical College Hospital, Chinese Academy of Medical Sciences and Peking Union Medical College, Beijing 100730, China; pumchzgykdxls@163.com (S.L.); xiaoyiding59@pumch.cn (Y.X.); hpharold@126.com (M.Z.); yunanze@pumch.cn (N.Y.); pumc_angzeng@sohu.com (A.Z.); 2Department of Neurological Surgery, Peking Union Medical College Hospital, Chinese Academy of Medical Sciences and Peking Union Medical College, Beijing 100730, China; wanghe_15@126.com

**Keywords:** keloid, laser speckle contrast imaging, deep learning, computer-aided workflow

## Abstract

A keloid results from abnormal wound healing, which has different blood perfusion and growth states among patients. Active monitoring and treatment of actively growing keloids at the initial stage can effectively inhibit keloid enlargement and has important medical and aesthetic implications. LSCI (laser speckle contrast imaging) has been developed to obtain the blood perfusion of the keloid and shows a high relationship with the severity and prognosis. However, the LSCI-based method requires manual annotation and evaluation of the keloid, which is time consuming. Although many studies have designed deep-learning networks for the detection and classification of skin lesions, there are still challenges to the assessment of keloid growth status, especially based on small samples. This retrospective study included 150 untreated keloid patients, intensity images, and blood perfusion images obtained from LSCI. A newly proposed workflow based on cascaded vision transformer architecture was proposed, reaching a dice coefficient value of 0.895 for keloid segmentation by 2% improvement, an error of 8.6 ± 5.4 perfusion units, and a relative error of 7.8% ± 6.6% for blood calculation, and an accuracy of 0.927 for growth state prediction by 1.4% improvement than baseline.

## 1. Introduction

Keloids are benign dermal fibroproliferative lesions that grow beyond the wound margin. Reddish, protuberant, solid, pruritic, and painful nodules or plaques on the skin are the common symptoms [1]. Despite advances in our understanding of wound healing and collagen metabolism, managing keloids remains a clinical challenge [2]. Timely discovery and treatment of progressive keloids can effectively inhibit the continued growth of keloids and improve both clinical prognosis and cosmetic effect.

The technique of laser speckle contrast imaging (LSCI) is based on an analysis of the reduction in speckle contrast, which is a real-time noncontact measurement [3]. Many studies have been proposed based on LSCI to evaluate the blood perfusion of keloids [4,5]. Those studies have shown that keloids have significantly higher blood-perfusion levels than adjacent and nonadjacent normal skin tissues of keloids, indicating the potential growth state.

However, LSCI-based blood-flow monitoring, a semi-automatic system, still has some drawbacks in clinical work [6]. The calculation of the blood perfusion of the keloid relies on the manual keloid boundary delineation, which is time consuming and subject to mislabeling. In particular, when detecting the blood flow of multiple keloids with irregular boundaries and irregular skin pigmentation, the phenomenon of missed labeling and mislabeling is more serious.

Nowadays, with the rapid development of deep learning, methods based on computer-vision modeling show a strong recognition ability for medical image analysis [7]. Notably, in dermatology and plastic surgery, deep learning is widely used to study skin with excellent performance, such as evaluating burn areas [8], differentiating multiple skin diseases [9], and diagnosing benign and malignant melanoma [10]. However, from the literature review, we find no relevant research on keloid evaluation and prognosis prediction.

This paper aims to construct a deep-learning-based workflow, integrate multiple deep-learning and machine-learning models, eliminate gaps between upstream and downstream tasks, and provide more application scenarios. The major research contributions are described as follows:(1)A cascaded vision transformer architecture is newly designed, which encodes and concatenates upstream and downstream features.(2)An automatic workflow for keloids’ clinical evaluation is proposed, including keloid segmentation, blood-perfusion calculation, and prognosis prediction.(3)An improved result is achieved by comparing it to traditional convolution neural networks.

## 2. Related Work

Few research studies are related to keloid evaluation, especially using the blood perfusion based on LSCI. This part mainly summarizes the most advanced and past works of skin lesions. Additionally, related deep-learning methods were introduced in this section.

### 2.1. Skin Lesion Segmentation

Recently, convolution neural networks (CNN) have been extensively excavated for skin-lesion segmentation, and most works followed a U-shaped structure [11]. For example, Yang et al. proposed a deep hybrid convolutional network derived from the UNet++ and EfficitentNet [12].

Instead of modifying the U-shaped structure, many works have tried to add attention blocks in the network. Dong et al. proposed a feedback attention network based on a context encoder network for skin lesion segmentation [13]. Tao et al. proposed another attention block named channel spatial fast attention-guided filter for the densely connected convolution network [14]. Wu et al. proposed an adaptive dual attention module to refine the extracted features for upsampling operations [15].

### 2.2. Skin Lesion Classification

Classification is the most common task in the computer vision area, and several techniques have been developed since ImageNet was brought up [16]. Afza et al. used hybrid deep-features selection and extreme learning machine for multiclass skin lesion classification [17]. Arshad et al. used ResNet50 and ResNet101 to extract features and the skewness-controlled SVR approach to select the best features for multiclass skin lesion classification [18]. Moldovanu et al. used an ensemble of machine-learning techniques instead of deep-learning methods [19]. Yao et al. proposed a multi-weight new loss function to classify skin lesions on an imbalanced small dataset [20].

### 2.3. Coupling Segmentation and Classification

Instead of independently segmenting and classifying, many methods have been developed to couple the segmentation and classification in one framework. Manzoor proposed a lightweight approach by CNN and features fusion [21]. Amin et al. used a threshold method to segment the lesion and used AlexNet and VGG16 to extract features to classify benign and malignant cancer [22]. Khan proposed an integrated framework including machine learning and CNN models for skin lesion detection and recognition [23]. 

By coupling segmentation and classification into the same framework, the classification accuracies were improved in the above studies.

### 2.4. Vision Transformer

The transformer was first used in natural language processing (NLP) and has become one of the best models [24]. Recently, studies have shown that by cutting images into patches, the transformer network, namely vision transformer (VIT) can outperform traditional CNNs in most computer-vision areas [25]. The vision transformer has become a state-of-the-art method with promising performance. Cao et al. adopted a pyramid transformer layer and constructed a Global and Local Inter-pixel correlation learning network for skin lesion segmentation [26]. Wu et al. proposed a FAT-Net, which integrated an extra transformer branch to capture long-range dependencies and global context information [27]. However, vision transformer (VIT) requires a large training dataset, which greatly limited application in medical image analysis with relatively small datasets.

## 3. Materials and Methods

### 3.1. Participant

To develop the deep-learning-based workflow, we retrospectively enrolled patients diagnosed with keloids in Peking Union Medical College Hospital from 2019 to 2021. The inclusion criteria were (1) patients with at least one keloid; (2) patients with saved black and white intensity images and color blood-perfusion images from LSCI; (3) patients without prior treatment; (4) patients without systemic disease; (5) patients with follow-up data. After screening, 150 patients were included in this study, and the collected keloids were from 7 regions, including the back, chest, ear, face, hip, limb, and abdomen. This study (No: S-K196) was approved by the China Association for Ethical Studies.

### 3.2. Device and Manual Annotation

Keloid perfusion was assessed by LSCI (PeriCam PSI System^®^; Perimed, Järfälla, Sweden), and Image analysis was undertaken using the manufacturer’s software (PimSoft 1.2.2.0^®^; Perimed, Järfälla, Sweden). The software produced both a blood-perfusion image and an intensity image of the scanned area. We collected blood-perfusion images and intensity images of the included patients. The measurement of blood perfusion was expressed in perfusion units (PUs, mL/100 g/min). Two plastic surgeons manually annotated the intensity images using “labelme” software [28]. Each gross photograph was segmented into two tissues, background (nonkeloid) and keloid (keloid). The segmented images will be used to supervise the training of an automated segmentation model for keloids.

### 3.3. Establishment of the Automatic Segmentation Module

Studies have shown that the vision transformer is a strong inference network with dense bias inductions, which requires a large amount of training data. However, this study’s training data are really limited, and a well-pretrained weight is necessary before training. Instead of using the weight pretrained in the ImageNet directly, we adopt a pretraining method, namely Masked AutoEncoder (MAE) [29]. For each image, 75% of patches are masked as the input, and the training target is to reconstruct the original image. In this pretraining method, the model is trained to learn a rich hidden representation and infer complex, holistic reconstructions. We followed the official implementation with 1600 pretraining epochs.

The images are resized into 512 × 512 and then cut into patches with the size of 16 × 16. Therefore, a total of 1024 patches are generated for each image. A VIT with 12 layers was used as the encoder, and the upernet was used as the segmentation decoder [30]. Manual annotations were used to supervise the neural network to learn how to act as an expert and segment the keloid automatically. The model was programmed based on PyTorch, and a random vertical and horizontal flip, and a random rigid rotation of 15 degrees were employed as the data augmentation. The SGD optimizer was used for backpropagation, and the learning rate was exponentially attenuated from 0.02. With a total training of 100 epochs, the training process took two hours using one 3090 Nvidia GPU.

### 3.4. Establishment of the Blood-Perfusion Analysis Module

Though we could not implant the program inside the LSCI to obtain the original blood-flow images, considering that the LSCI will generate a heat map (from blue to red) of the blood-perfusion image, we can indirectly calculate the blood-perfusion value at each pixel according to the color. After aligning the color bar with the blood-perfusion value, we established a mapping function between the RGB of the image and the blood-perfusion value [31]. Further, based on the automatic segmentation results, we cropped the blood flow image in the keloid area and calculated its average blood-perfusion value. The automatically computed blood-perfusion value was compared with the origin blood-perfusion value obtained in LSCI to get the relative perfusion error.

### 3.5. Establishment of the Evaluation Module

Considering that the blood perfusion inside the keloid will reflect its intrinsic growth state and the keloid usually presents an uneven perfusion distribution, we did not directly use the mean blood-perfusion value to evaluate the growth state of the keloid. The blood-perfusion image was firstly masked by the segmentation result to remove unnecessary blood perfusion and then resized to 512 × 512. Similarly, the blood-perfusion image was cut into 1024 patches with the size of 16 × 16. A vision transformer was used to encode the blood-perfusion images.

To remove unnecessary regions and save memory costs, only patches with blood perfusion were fed into the transformer. After perfusion encoding, the perfusion features were concatenated with the intensity features from the intensity encoder. Finally, a prediction decoder with 4 transformer layers decoded the composited feature and output a three-classification prediction, namely regressive, stable, and progressive. Patients reported three types of keloid growth stages based on whether the keloid grew smaller, stayed the same size, or grew larger over the previous year [5]. The training was performed in an automatic augmentation manner with random erasing of 25% of pixels [32,33]. The initial learning rate was set to 0.001, and attenuated by 1/2 at 50, 100, and 200 epochs. This training process took about two hours on 3090 Nvidia GPU with a total of 400 training epochs.

### 3.6. Establishment of the Workflow and Evaluation Matrix

As shown in Figure 1, the purpose of this study was to improve the automation and diagnostic ability during the diagnosis and treatment process of keloids in clinical practice. In this study, three diagnosis modules were designed: an automatic segmentation module, a blood-perfusion analysis module, and an evaluation module. The whole workflow was a semi-automatic diagnosis procedure with a cascaded vision transformer structure. First, it is necessary to manually obtain the unlabeled intensity images and blood-perfusion images of the patient and then put them into the automatic segmentation module to segment the keloid. The segmentation result and the original blood-perfusion image were put into the blood-perfusion analysis module to obtain the average blood-perfusion and cropped blood-perfusion image. Finally, the cropped blood-perfusion image, together with previously encoded intensity features, were put into the evaluation module to evaluate the keloid growth state.

The modules designed in this paper were trained and validated independently using 5-fold cross-validation. For each module, we selected an independent evaluation function.

For the automatic segmentation module, we used the DICE value as the evaluation criteria. The higher the DICE value, the more accurate automatic segmentation results.
$$\mathrm{DICE}=\frac{2\left|A\cap B\right|}{\left|A\right|+\left|B\right|}$$
where A and B represent the features to be evaluated from the manual segmentations and deep-learning results, respectively.

For the difference in blood perfusion, we used the perfusion blood error and the relative blood-perfusion difference to evaluate.
perfusion blood error=|A−B|
$$\mathrm{relative}\;\mathrm{perfusion}\;\mathrm{blood}\;\mathrm{error}=\frac{\left|A-B\right|}{\max\left(\left|A\right|,\left|B\right|\right)}\times 100\%$$
where A and B represent the blood-perfusion value obtained from the original LSCI and the blood-perfusion module, respectively.

For the final evaluation analysis, we reported sensitivity, specificity, Youden index, and accuracy for each category.

## 4. Results

### 4.1. Study Participants

In all, 150 keloids were enrolled in this study. As shown in Table 1, there were 75 men and 75 women with a mean age of 30.6 ± 11.1 years, a mean keloid duration of 7.1 ± 3.9, and a mean blood-perfusion of 129.9 ± 41.0. There were 49 (32.7%) keloids in the regressive stage, 37 (24.7%) in the stable stage, and 64 (42.7%) in the progressive stage. Blood perfusion varied greatly at different locations. Keloids on the face had the maximum mean blood-perfusion of 182.8 ± 23.0 PU, while keloids on the hip had the minimum mean blood-perfusion of 103.0 ± 40.2 PU.

### 4.2. Segmentation Module

After being trained in a five-fold training manner, the mean DICE value was calculated over five training processes. The proposed method finally achieved the mean DICE value of 0.895. Examples of segmentation can be found in Figure 2. The first column shows the original intensity image, the second column shows the manual annotation, and the third column shows the automatic segmentation. We can find that automatic segmentation has a smoother and more regular boundary than manual segmentation.

Ablation studies (Table 2 segmentation) were performed among architectures and pretraining methods. Resnet and hrnet were listed as the baseline methods [34,35]. When the pretraining weights were abandoned, the traditional CNNs outperformed VIT by about 11% DICE value (HRnet-c1 0.671 vs. VIT 0.562). However, when using ImageNet pretraining weights, the performance among CNNs and VIT were close (HRnet-c1 0.875 vs. VIT 0.870). Specifically, when using MAE as the pretraining method, the performance of VIT was further improved with the DICE value of 0.895 and outperformed CNNs by 2%. 

### 4.3. Blood-Perfusion Module

Blood perfusions were calculated based on the automatic segmentation results and manually acquired blood-perfusion images. The cropped image is shown in the last column in Figure 2, eliminating the effect of skin pigmentation. The proposed blood-perfusion module showed high accuracy, with a small mean perfusion error of 8.6 ± 5.4 PU and a relative perfusion error of 7.8% ± 6.6%.

### 4.4. Evaluation Module

After masking the blood-perfusion image with the segmentation and cutting images into patches, the features were encoded with the blood features. Both blood features and intensity features were used for the final prediction. Results showed that the automatic evaluation module achieved high prediction accuracies. Table 3 reported the sensitivities, specificities, Youden index, and accuracies. The sensitivities of the three stages were 0.936, 0.892, and 0.939. The specificities of the three stages were 0.961, 0.965, and 0.964, respectively. The evaluation module showed a mean accuracy of 0.927.

Ablation studies (Table 2 prediction) showed that deep-learning-based methods could achieve accuracy of 0.89 even without pretraining, indicating the growth state is a relatively easy task to predict. Similarly, CNNs outperformed VIT without pretraining (Resnet101 0.893 vs. VIT 0.887). After using the patch selection and concatenation, VIT outperformed CNNs by 1.4% (Resnet101 0.913 vs. VIT 0.927).

## 5. Discussion

Many studies have shown that the application of LSCI can more accurately assess the growth state of the keloid, and high blood-perfusion often indicates that the keloid is more active [4,5,36]. The characteristics of the patients we enrolled (Table 1) showed that the blood flow of keloids at different sites had significant differences, which is consistent with previous studies [4]. These studies have established keloid evaluation methods using blood perfusion to avoid the subjective error caused by the traditional VSS (Vancouver Scar Scale) or VAS (Visual Analogue Scale) score. Our research was a structural keloid evaluation method established based on LSCI.

High blood-perfusion has been linked to microangiogenesis in damaged skin in previous investigations [37]. However, this conclusion appears to contradict that of other studies, which claimed that the keloid’s blood flow was impaired [38,39,40]. The perfusion level in keloids was previously estimated based on a single factor such as vascular density, transverse area, or shape, whereas our current work used LSCI to measure perfusion. Blood flow within the keloid evaluated by Laser Doppler Flowmetry (LDF) had a good association with the overall scores of a validated grading system encompassing redness, elevation, hardness, itch, and pain, according to Olimpia Timar-Banu et al. [41]. However, in our study, LSCI has a higher picture quality, repeatability, scanning efficiency, and lower variability than LDF because it does not need to be in touch with the targeted skin [42].

Artificial intelligence has been widely used in dermatology, such as psoriasis area evaluation, skin disease diagnosis, and melanoma diagnosis [9,10,43]. However, there is still no study on the AI-assisted diagnosis and treatment process for keloids. This study aims to simplify the clinical diagnosis and treatment process, assist clinical decision-making, and propose a highly automated workflow. Some patients have multiple and irregular keloids, which significantly prolongs manual delineation time and leads to mislabeling and missed labeling. The use of an automatic keloid segmentation module can considerably shorten the process and avoid human error. Through the workflow designed in this study, the computation time of one patient can be reduced to less than one second.

Previous research has shown intensity images and blood-perfusion images based on LSCI may not be compatible, and keloids with regular borders can exhibit heterogeneous blood-perfusion [44,45]. Studies have shown that the performance of LSCI in normal skin can also be affected by skin pigmentation, which brings interference to the direct application of LSCI images when predicting scar growth state [46]. In this study, based on accurate keloid segmentation, the LSCI image was cropped to the keloid region, thus eliminating the effect that skin pigmentation brings to the blood-perfusion images. On the other hand, based on accurate keloid segmentation, the evaluation module can focus more attention on the internal inhomogeneity of the keloid and make a precise evaluation of the current growth state.

**Ablation studies:** The proposed cascaded transformer architecture was compared to the traditional CNNs. Our results showed that CNNs outperformed VIT in segmentation and classification when pretraining weights were abandoned. However, the VIT performed similarly with CNNs when networks were well initialized. We believed the above observation was due to the training process of VIT requiring a large amount of data, while this study only included 150 samples. Using the MAE pretraining method, we found the VIT could be well initialized with dataset-specific hidden representation, which significantly improved the result in the small dataset. The proposed patch selection and feature concatenation method could integrate upstream and downstream features and save more than 90% memory cost.

**Strengths and limitations****:** This study used unlabeled intensity images and blood-perfusion images based on LSCI as inputs, which were easy to obtain without additional manual intervention. After being trained with limited training samples and manual annotations, the deep-learning model does not require excessive human intervention during testing. Therefore, the workflow constructed in this paper requires low hardware, labor, and time costs, which is conducive to research and promotion in more clinical scenarios. At the same time, manual delineation often has rough boundaries and cannot avoid subjective bias. However, the proposed workflow has robust reproducibility while avoiding human errors and segmenting keloids with regular boundaries. Besides, the deep-learning model will be self-updated with increasing labeling quality and training samples in the future. In contrast, the improvement of human work is very limited after reaching saturation of the learning curve.

This study has some limitations. Among them, retrospective studies are the most significant research obstacle. Due to the lack of prospective design, none of the enrolled patients underwent keloid excision surgery. However, the postoperative keloid is more irregular, and our model may not accurately segment the postoperative keloids, which leads to the inability to evaluate the growth state of the keloid correctly. In future studies, we will further collect postoperative patient data to improve the model’s generalization. Additionally, the workflow designed in this study only covers the primary feature extraction, such as blood-perfusion value and intrinsic growth state. To predict other indicators, including pain and pruritus, it is still necessary to supplement more data to expand application scenarios.

## 6. Conclusions

This paper proposed a workflow with a cascaded transformer architecture for evaluating keloid states based on LSCI, which contained three modules: an automatic segmentation module, a blood-perfusion analysis module, and an evaluation module. We used the automatic segmentation module to segment and located the keloid, reaching a DICE value of 0.895. Based on the automated segmentation results and the blood-perfusion images, we automatically cropped the image into the keloid blood-perfusion area and calculated its average blood-perfusion value, which achieved an error of 8.6 ± 5.4 PU and a relative error of 7.8% ± 6.6%. Further, we cut the blood-perfusion images into patches and fed them into the evaluation module to predict the growth state, and finally, we achieved an average accuracy of 0.927. The workflow we designed integrates segmentation, analysis, and evaluation, which can assist and simplify the assessment of the keloid in future clinical work.

## Figures and Tables

**Figure 1 jpm-12-00981-f001:**
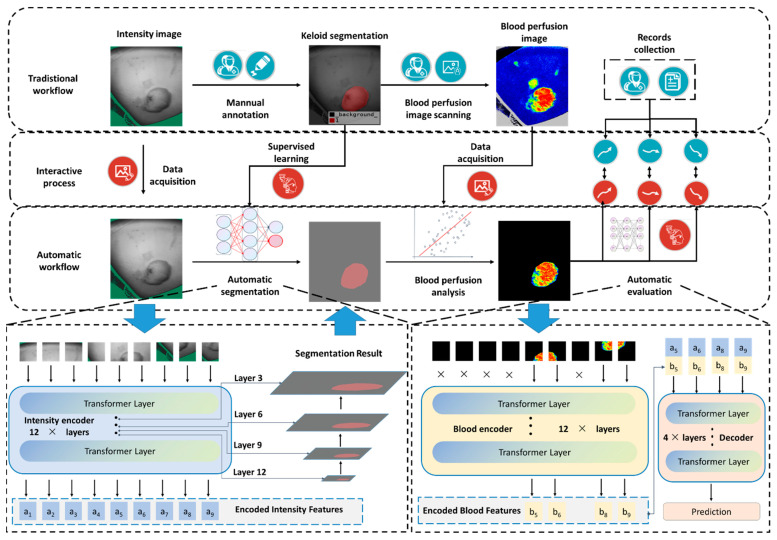
Proposed workflow for AI-assisted keloid segmentation and evaluation. We proposed a workflow with a cascaded vision transformer architecture for evaluating keloid states based on LSCI (laser speckle contrast imaging), which contained three modules: an automatic segmentation module, a blood-perfusion analysis module, and an evaluation module. The automatic segmentation module was used to segment and located the keloid, the blood-perfusion analysis module was used to crop the blood-perfusion image to the keloid area, and the evaluation module was used to evaluate the keloid growth states (regressive, stable, and progressive).

**Figure 2 jpm-12-00981-f002:**
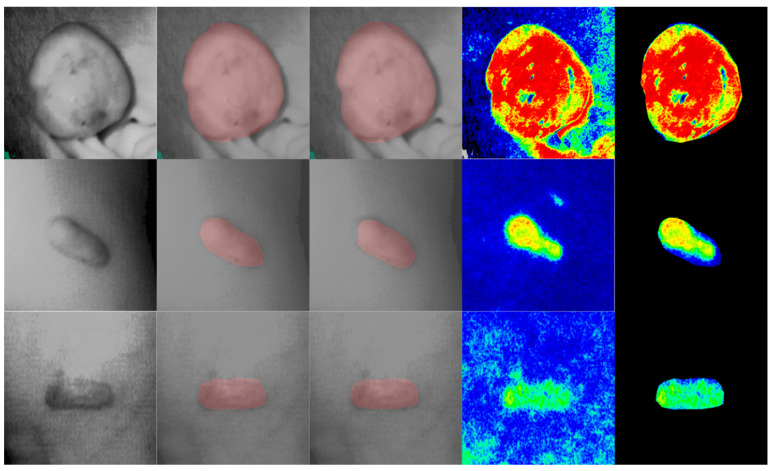
Segmentation and cropping results of proposed modules. We showed three examples in this figure. The first column, original intensity images; the second column, manual annotations; the third column, automatic segmentations; the fourth column, original blood-perfusion images; the final column, the cropped blood-perfusion images. The first row showed the keloid in the progressive stage; the middle row showed the keloid in the stable stage; the last row showed the keloid in the regressive stage.

**Table 1 jpm-12-00981-t001:** Demographic characteristics.

Location	N	Male	Female	Age	Duration	Perfusion	Regressive	Stable	Progressive
Back	34	18	16	33.6 ± 11.4	7.6 ± 3.9	127.6 ± 43.7	10	9	15
Chest	63	29	34	29.5 ± 12.3	6.8 ± 4.1	135.8 ± 35.6	14	16	33
Ear	8	4	4	26.9 ± 10.3	6.1 ± 5.0	157.8 ± 41.7	1	3	4
Face	6	4	2	27.8 ± 3.7	9.7 ± 4.1	182.8 ± 23.0	0	1	5
Hip	9	5	4	34.2 ± 8.8	6.0 ± 3.3	103.0 ± 40.2	7	1	1
Limb	18	8	10	30.3 ± 10.4	6.7 ± 4.0	105.2 ± 38.4	12	3	3
Abdomen	12	7	5	29.3 ± 8.2	8.0 ± 4.2	118.5 ± 32.7	5	4	3
All	150	75	75	30.6 ± 11.1	7.1 ± 3.9	129.9 ± 41.0	49	37	64

**Table 2 jpm-12-00981-t002:** Ablation study.

	Segmentation		Prediction		
Method	Pretrain	DICE	Method	Pretrain	Accuracy
Resnet50-upernet	None	0.651	Resnet50	None	0.893
HRnet-c1	None	0.671	Resnet101	None	0.893
Resnet50-upernet	ImageNet	0.861	Resnet50	ImageNet	0.907
HRnet-c1	ImageNet	0.875	Resnet101	ImageNet	0.913
VIT-base-upernet	None	0.562	cascade-VIT	None	0.887
VIT-base-upernet	ImageNet	0.870 (−0.005)	cascade-VIT	ImageNet	0.913 (+0)
VIT-base-upernet	MAE	**0.895** (+0.020)	+patch selection	ImageNet	**0.927** (+0.014)

Note: the bold texts show the best result for each task.

**Table 3 jpm-12-00981-t003:** Results of the evaluation module.

	Regressive	Stable	Progressive	All
Sensitivity	0.936	0.892	0.939	
Specificity	0.961	0.965	0.964	
Youden	0.897	0.856	0.904	
Accuracy	0.953	0.947	0.953	0.927

## Data Availability

The data and code used to support the findings of this study are available from the corresponding author on request.

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
