# Peer review of "A Workflow for Computer-Aided Evaluation of Keloid Based on Laser Speckle Contrast Imaging and Deep Learning"

_jpm, 2022, doi:10.3390/jpm12060981_

Round 1

Reviewer 1 Report

The paper described the development of AI for keloids diagnostic and its efficay.   A nd provided the new ideas for diagnostic of keloids with AI.  only few articles are published on this topic 

Author Response

Thanks!

Reviewer 2 Report

This work is not enough contribution and innovation. However, the problem statement and motivation could be stronger or more clearly highlighted.

1.      The existing literature should be classified and systematically reviewed, instead of being independently introduced one-by-one.

2.      The abstract is too general and not prepared objectively. It should briefly highlight the paper's novelty as what is the main problem, how has it been resolved and where the novelty lies?

3.      For better readability, the authors may expand the abbreviations at every first occurrence.

4.      The author should provide only relevant information related to this paper and reserve more space for the proposed framework.

5.      However, the author should compare the proposed algorithm with other recent works or provide a discussion. Otherwise, it's hard for the reader to identify the novelty and contribution of this work.

6.      The descriptions given in this proposed scheme are not sufficient that this manuscript only adopted a variety of existing methods to complete the experiment where there are no strong hypothesis and methodical theoretical arguments. Therefore, the reviewer considers that this paper needs more works.

The algorithm presented has not any novelty.

7.      The related works section is very short and no benefits from it. I suggest increasing the number of studies and add a new discussion there to show the advantage.  Following can be included in related work

a.      Developed Newton-Raphson based deep features selection framework for skin lesion recognition

b.      Integrated design of deep features fusion for localization and classification of skin cancer

c.      An integrated framework of skin lesion detection and recognition through saliency method and optimal deep neural network features selection

d.      A Lightweight Approach for Skin Lesion Detection Through Optimal Features Fusion

e.      Machine-learning-scheme to detect choroidal-neovascularization in retinal OCT image

8.      The manuscript is not well organized. The introduction section must introduce the status and motivation of this work and summarize with a paragraph about this paper.

Reviewer 3 Report

The work describes the use of deep learning to help in identifying keloid stages which is novel and beneficial for the medical application. Some minor comments here for improvements:

Minor formatting and grammar errors are seen.

L61: Has the ethical approval been granted? If so, please provide the approval number and evidence if necessary.

L90: The blood perfusion value was calculated based on the image. Few things to address here that may affect the estimation:

-Has such relevancy been reported? Provide appropriate evidences

-Has the author considered the different skin colour/pigmentation?

-As keloid is a 3D structure, the edges of the keloid in the photo tend to have a darker pixel. Can this affect the calculation?

Fig 3: Were these the keloid of different stages? if yes, may name them so the readers can get an idea; if not, provide them

Round 2

Reviewer 2 Report

The paper can be accepted.